# Searching for Better Spatio-temporal Alignment in Few-Shot Action Recognition

**Yichao Cao**[1*], **Xiu Su**[2*], **Qingfei Tang**[3], **Shan You**[4], **Xiaobo Lu**[1], **Chang Xu**[2†]
[1]School of Automation, Southeast University,
[2]School of Computer Science, Faculty of Engineering, The University of Sydney,
[3]Enbo Technology Co.,Ltd., China,
[4]SenseTime Research
caoyichao@seu.edu.cn, xisu5992@uni.sydney.edu.au, qingfeitang@gmail.com
youshan@sensetime.com, xblu@seu.edu.cn, c.xu@sydney.edu.au

## Abstract

Spatio-Temporal feature matching and alignment are essential for few-shot action recognition as they determine the coherence and effectiveness of the temporal patterns. Nevertheless, this process could be not reliable, especially when dealing with complex video scenarios. In this paper, we propose to improve the performance of matching and alignment from the end-to-end design of models. Our solution comes at two-folds. First, we encourage to enhance the extracted Spatio-Temporal representations from few-shot videos in the perspective of architectures. With this aim, we propose a specialized transformer search method for videos, thus the spatial and temporal attention can be well-organized and optimized for stronger feature representations. Second, we also design an efficient non-parametric spatio-temporal prototype alignment strategy to better handle the high variability of motion. In particular, a query-specific class prototype will be generated for each query sample and category, which can better match query sequences against all support sequences. By doing so, our method SST enjoys significant superiority over the benchmark UCF101 and HMDB51 datasets. For example, with no pretraining, our method achieves 17.1% Top-1 accuracy improvement than the baseline TRX on UCF101 5-way 1-shot setting but with only 3x fewer FLOPs.

## 1 Introduction

One of the most important tasks in video understanding is to understand human actions, which is one of the representative tasks for video understanding [54]. In recent years, video understanding and action recognition have been witnessing remarkable progress with the emergence of high-quality large-scale video datasets [5, 15, 10]. However, this success relies quite heavily on a large amount of manually labeled samples. The dataset annotation process is laborious, cumbersome, and time-consuming and restricts practical algorithm applications. Therefore, how to classify unseen action classes with extremely few annotated samples gives rise to the investigation of few-shot (FS) action recognition [52, 4].

Compared with the image task [49, 46, 48, 8, 50, 45], introducing temporal dimension makes the video task more complicated. For example, actions in videos are often performed at various speeds and occur at different time points. Further, action recognition usually needs to integrate multiple distinctive sub-action information to generate corresponding temporal representations, which are

---

[*]Equal contribution.
[†]Corresponding author.

36th Conference on Neural Information Processing Systems (NeurIPS 2022).

used for subsequent spatio-temporal feature matching. To address the few-shot action classification problem, most existing studies [52, 4, 37, 3, 32, 39] use metric-based meta-learning framework to implement similarity comparison between query (test) and support (reference) videos. It first maps the input video to an embedding space via representation learning and then achieves the distance metric for comparing video similarities in an episodic task. These approaches follow the episodic training to meta-learn a backbone network [52] or temporal relational module [4, 3, 39]. However, we argue that these methods underestimate the importance of spatio-temporal representation, which is crucial for the basic concept of few-shot action classification. And, there are some concerns about existing temporal relational strategies [39] on spatial features: (a) insufficient sparse sampling frames may limit the capacity for long-term temporal models, and (b) increasing the number of sampling frames will lead to the complexity of combination and matching strategy.

Inspired by these works, we present a novel approach to explore the prime architectures as to sequence order of temporal attention and spatial attention blocks for the few-shot action recognition. To boost the search, we leverage a pre-defined transformer space that considers both temporal attention and spatial attention, and propose a space shrinking strategy with the analysis of collected training losses during the search to explore the emphasis of spatial and temporal based information in different stages for the superior architectures in terms of accuracy and budgets. Finally, we propose a nonparametric spatio-temporal prototype alignment method to encourage each query video to match all reference videos in the support set for long-term temporal models.

The main contribution of this paper is summarized as follows:

- To explore the possibility of the optimal video understanding architecture for few-shot action recognition, we incorporate spatial and temporal attention elements in our transformer space. It allows the model to spontaneously choose and focus on different types of information at various stages, which is crucial for obtaining better spatio-temporal representation.

- We introduce a transformer space shrinking strategy to adaptively evolve Transformer space to speed up the supernet training and structure search. Based on this strategy, the time and cost for video understanding architecture search are drastically decreased.

- We propose a more efficient and general spatio-temporal prototype alignment method, which can be conveniently adapted to arbitrary length video matching. Our method breaks the limits of frames of standard relational modules for few-shot action recognition and achieves better capacity for long-term temporal models.

## 2   Related Work

**Few-shot Image Classification**. Few-shot learning [13, 25, 26] aims to learn novel classes with only a small number of labeled samples. One of the fundamental tasks in FSL is few-shot image classification [28, 18], which is preferable to using a combination of representation learning and a metric-based classifier. The terminology "$n$-way $k$-shot" means that models are used to classify $n$ distinct classes with only $k$ reference samples per class. This process is also called "episode" [42]. Recent works often average the representation of a support class to calculate a more robust class prototype [18, 12]. CrossTransformers [11] employs spatial alignment between query- and support-set images to improve generalization for unseen classes. Matching-based methods [14, 30, 38, 43] tend to learn a deep distance metric or similarity functions to compare examples [7, 9, 20, 40].

**Few-shot Video Action Recognition**. Recent attempts [19, 54, 23, 51, 41, 56, 5] focus more on temporal alignment [55], which aims to align two video sequences in temporal dimension. Notably, [4] explicitly learns distance measure and representation independent of non-linear temporal variations in video sequences. [53] proposes a prototype-centered contrastive learning loss and a hybrid attentive learning mechanism to minimize the negative impacts of outliers and promote class separation in few-shot action recognition task. [22] designs temporal transform module to handle action misalignment, which consists of two parts: a localization network and a temporal affine transformation. [24] formulates the distance between video sequences as transportation cost. Temporal Relational CrossTransformer (TRX) [27] operates over a different number of frames for higher-ordered temporal model and query-specific class prototypes construction. However, this method combines a small number of frames that may have limited capacity for modeling long-term dependencies. In addition, as the number of input frames grows, the number of combinations increases sharply, which makes the processing of temporal relation module more complicated.

**Neural Architecture Search (NAS)**. NAS [36, 33, 44] aims to automatically learn/search for better neural network architectures. Earlier, MetaQNN [1] is proposed to search high-performing CNN architectures via reinforcement learning. [29] proves that evolutionary algorithms are capable of constructing large, accurate networks in a huge search space. Single Path [17] proposes to train each architecture alternately with a two-stage search schema. AutoFormer [6] first adopted the one-shot NAS framework for the ViT based architecture search.

# 3   Method

## 3.1   Problem Formulation

The goal of few-shot video action classification is to classify an unlabelled query video into several new classes with only a few labeled video samples, referred to as the 'support set'. The dataset is separated into a training set $\mathcal{D}_{tr} = \{(x_i, y_i) \mid y_i \in \mathcal{C}_{train}\}$, a validation set $\mathcal{D}_{val} = \{(x_i, y_i) \mid y_i \in \mathcal{C}_{validation}\}$, and a test set $\mathcal{D}_{test} = \{(x_i, y_i) \mid y_i \in \mathcal{C}_{test}\}$, where the validation set is split from the training set with $1/10$ samples, $x_i$ is the $i$-th video sample with class label $y_i$. Different from the general supervised learning, few-shot training and test categories are disjoint, i.e. $\mathcal{C}_{train} \cap \mathcal{C}_{test} = \emptyset$. To make training more faithful to the testing method, we follow episodic training [27][47] to train the model. In each episodic task, training samples are randomly sampled from the training set in a $n$-way $k$-shot setting. Concretely, a support set $\mathbb{S}$ and a query set $\mathcal{Q}$ are sampled from $\mathcal{D}_{tr}$ as $\mathbb{S}, \mathcal{Q} = \{(x_i, y_i) \mid y_i \in \mathcal{C}_{support}\}$, where $\mathcal{C}_{support} \subseteq \mathcal{C}_{train}$ is the $n$ selected classes. Compared with few-shot image classification, the introduction of the time dimension in video classification demands the model to be able to learn temporal patterns with limited sample size. Next, we describe our proposed method for few-shot video action recognition.

## 3.2   NAS for Few-shot Video Action Recognition

For a supernet [34] $\mathcal{N}$ with weight $\mathcal{W}$ and Transformer space $\mathcal{A}$, each subnetwork (subnet) $\boldsymbol{a}$ inherits its weight $\boldsymbol{w_a}$ from $\mathcal{W}$. The optimization and search problem in our supernet training stage can be formulated as:

$$\boldsymbol{a}^* = \operatorname*{arg\,max}_{\boldsymbol{a} \in \mathcal{A}} \ \text{ACC}_{val}(\boldsymbol{a}, \mathcal{W}_{\mathcal{A}}^*(\boldsymbol{a}), \mathcal{D}_{val}) \tag{1}$$

$$\text{s.t.} \ \mathcal{W}_{\mathcal{A}}^* = \operatorname*{arg\,min}_{\mathcal{W}_{\mathcal{A}}} \mathcal{L}_{ce}\left(\boldsymbol{w_a}; \mathcal{N}_{\mathcal{A}}, \boldsymbol{a}, \mathcal{D}_{tr}\right) \tag{2}$$

where $\mathcal{W}_{\mathcal{A}}^*$ is a set of trained weights for the whole Transformer space $\mathcal{A}$, $\mathcal{L}_{ce}$ and $\text{ACC}_{val}$ denote the cross-entropy loss and the accuracy for each subnet $\boldsymbol{a}$ in the episodic task, respectively.

We build our overall Transformer space $\mathcal{A}$ with $4.74 \times 10^{18}$ subnets as in Table 1. Since the video understanding model may need to adjust the layout of temporal and spatial attention in different layers for better temporal model ability, we design independent Space "SAB" and Time Attention Blocks "TAB" for our Transformer space. Although the video features are transformed by TAB and SAB, the output feature dimension is unchanged. To avoid the model being overly complex, We incorporate the dimension downsample block, named "Spatial Downsampling Block (SDB)" in Table 1 as in [16]. The core operation of this block is to use the down sampling in spatial self-attention, which maps an input tensor of size $(C, W, H)$ to an output tensor of size $(C', W/2, H/2)$ with $C' > C$. Due to the change in scale, we can easily control the complexity of the model in different stages. We present more details about our Transformer space in section 3.5.

## 3.3   Transformer Space Shrinking

In the previous subsection, we design a large Transformer [35] space covering a variety of operations to explore the optimal network structure. However, this also causes some problems: 1) too large Transformer space increases the difficulty of training and optimization of supernet, 2) redundant operations in Transformer space increase the difficulty of structure search. How to design a variety of Transformer spaces for different parts is another challenging task. To this end, we propose a Transformer space evolving strategy. Towards a given Transformer space, the unreasonable space parts are automatically shrunk, and the whole space gradually converges to a more compact state.

To evaluate each operation in Transformer space, we define score $\mathcal{S}(i, j)$ as the evaluation metric for $j$-th operation in $i$-th layer. Formally, any subnet $\boldsymbol{a} \in \mathcal{A}$ can be expressed as the result of

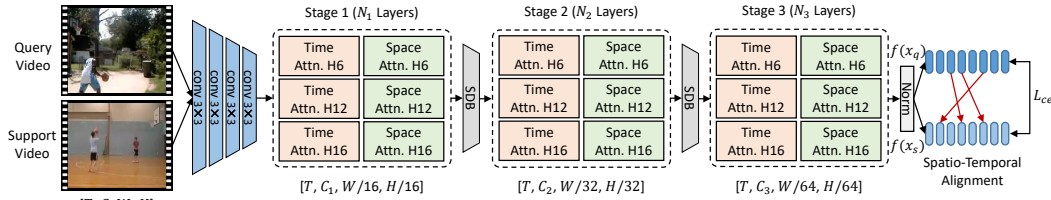

Figure 1: Overview of the neural module to be searched. The architecture of the network is determined by different operations per layers. There are three main stages in which space and time attention operations can be searched and selected.

---

**Algorithm 1:** Training supernet with Transformer space shrinking

---

**Input:** Supernet $\mathcal{N}$ with weight $\mathcal{W}$ and Transformer space $\mathcal{A}$, maximum training epochs $\mathcal{T}$, warm up epochs $\mathcal{T_w}$, shrink epochs $\mathcal{P}$, score threshold $Thr$ and shrink percentage $\mathcal{K}$.

Init $\tau = 0$;
**while** $\tau \leq \mathcal{T}$ **do**
    randomly sample subnets from supernet $\mathcal{N}$;
    train one-shot supernet with Transformer space $\mathcal{A}$;
    **if** $\tau \geq \mathcal{T_w}$ **then**
        record the loss and FLOPs of the subnet $\boldsymbol{a}_{i,j}^{o}$ for each operation $\boldsymbol{o}_{i,j}$;
        **if** $\tau\%\mathcal{P} = 0$ **then**
            calculate the $\mathcal{S}(i,j)$ for each operation with Eq. (6) and Eq. (7);
            $\mathcal{A} \leftarrow \mathcal{A}.\text{Shrink}(\mathcal{K}, Thr)$
**end**
**Output:** The optimized weight $\mathcal{W}_{\mathcal{A}}^{*}$ for supernet $\mathcal{N}$, and the shrunk Transformer space $\mathcal{A}$.

---

layer by layer operation stacking. The subnet $\boldsymbol{a}$ can be represented as $\boldsymbol{a} = \bigcup_i \sum_j \mathbf{1}_j^i \boldsymbol{o}_{i,j}$, where $\mathbf{1}_j^i \in \{0, 1\}$ denotes indicator function, which indicates whether $\boldsymbol{o}_{i,j}$ is selected in $\boldsymbol{a}$. In each layer of supernet, subnet selects only one operation. Thus, for the indicator function $\mathbf{1}_j^i \in \{0, 1\}$ in $i - th$ layer, the sum of the indicator functions is 1: $\sum_j \mathbf{1}_j^i = 1$. The training loss for subnet $\boldsymbol{a}$ can also be expressed as

$$\mathcal{L}oss(a) = \mathcal{L}oss \left( \bigcup_i \sum_j \mathbf{1}_j^i \boldsymbol{o}_{i,j} \right) = \mathcal{L}_{ce} \left( \boldsymbol{w_a}; \mathcal{N_A}, \boldsymbol{a}, \mathcal{D}_{tr} \right) \tag{3}$$

To represent the bi-directional relationship between operation $\boldsymbol{o}_{i,j}$ and subnet $\boldsymbol{a}$, we denote the subnet which includes $\boldsymbol{o}_{i,j}$ by $\boldsymbol{a}_{i,j}^o$. For a subnet $\boldsymbol{a}$ with $\boldsymbol{o}_{i,j}$, it is with huge variance to directly measure the contribution of $\boldsymbol{o}_{i,j}$ to the performance of $\boldsymbol{a}$. However, the same operation will appear in many subnets, and we can transverse the search space and leverage the performance expectation of all these subnets $\boldsymbol{a}_{i,j}^o$ to evaluate the score of $\boldsymbol{o}_{i,j}$. Thus, we can obtain the relationship between score and training loss:

$$\mathcal{S}(i,j) = \mathcal{S}\left( \boldsymbol{o}_{i,j} \right) \propto \mathbb{E}_{\boldsymbol{a} \in U(\mathcal{A}), \boldsymbol{o}_{i,j} \in \boldsymbol{a}} \left[\!\!\left[ \mathcal{L}_{ce} \left( \boldsymbol{w}_{\boldsymbol{a}_{i,j}^o}; \mathcal{N_A}, \boldsymbol{a}_{i,j}^o, \mathcal{D}_{tr} \right) \right]\!\!\right] \tag{4}$$

Considering that different operations have distinct effects on the computing budgets of subnets, it is often desirable to limit the computing budgets of subnets in practical scenarios. However, with the sufficient and same amount of data, those models with big computing budgets may have more advantages in persuing good results. If left unconstrained, the operation scores can be as

$$\mathcal{S}\left( \boldsymbol{o}_{i,m} \right) > \mathcal{S}\left( \boldsymbol{o}_{i,n} \right), if \; \mathbb{E}_{\boldsymbol{a} \in U(\mathcal{A}), \boldsymbol{o}_{i,j} \in \boldsymbol{a}} \left[\!\!\left[ \mathcal{B}\left( \boldsymbol{a}_{i,m}^o \right) \right]\!\!\right] < \mathbb{E}_{\boldsymbol{a} \in U(\mathcal{A}), \boldsymbol{o}_{i,j} \in \boldsymbol{a}} \left[\!\!\left[ \mathcal{B}\left( \boldsymbol{a}_{i,n}^o \right) \right]\!\!\right] \tag{5}$$

where $\mathcal{B}(\boldsymbol{a}_{i,j}^o)$ denotes the computing budget of subnet $\boldsymbol{a}_{i,j}^o$. To address this problem, we propose a modified approach considering the impact on the computing budget of subnets. We update Eq. (4) as

$$\mathcal{S}(i,j) = \mathcal{S}\left( \boldsymbol{o}_{i,j} \right) = \mathbb{E}_{\boldsymbol{a} \in U(\mathcal{A}), \boldsymbol{o}_{i,j} \in \boldsymbol{a}} \left[\!\!\left[ \mathcal{L}_{ce} \left( \boldsymbol{w}_{\boldsymbol{a}_{i,j}^o}; \mathcal{N_A}, \boldsymbol{a}_{i,j}^o, \mathcal{D}_{tr} \right) \right]\!\!\right] + \delta \left( \overline{\mathcal{B}_{i,j}^o} \right) \tag{6}$$

where $\delta(\cdot)$ is a correction term related to the statistical expectation for subnet $\boldsymbol{a}_{i,j}^o$, $\overline{\mathcal{B}_{i,j}^o}$ denotes the statistical expectation for subnet $\boldsymbol{a}_{i,j}^o$. We hope to correct the impact of different operations on the

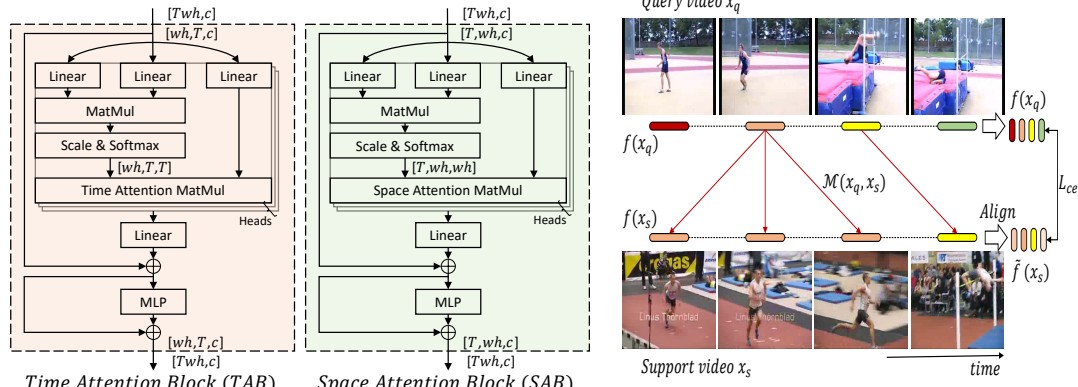

Figure 2: The essential operation of stack layer.

Figure 3: Spatio-temporal Prototype Alignment.

subnet FLOPs by $\delta(\cdot)$, so as to accurately evaluate the relative performance of different opertions by $\mathcal{S}(i,j)$. Then, the correction term can be defined as

$$\delta\left(\overline{\mathcal{B}_{i,j}^o}\right) = -\mathbb{E}_{\boldsymbol{a}\in U(\mathcal{A}),\mathcal{B}(\boldsymbol{a})=\overline{\mathcal{B}_{i,j}^o}}\left[\mathcal{L}_{ce}\left(\boldsymbol{w_a};\mathcal{N}_{\mathcal{A}},\boldsymbol{a},\mathcal{D}_{tr}\right)\right] \tag{7}$$

Those high-scoring (low performance) operations that deviate from the average score by more than a certain threshold will be dropped.

Specifically, we make statistics for all the operations after each $\mathcal{P}$ epochs during supernet training. We discard low-performing operations that deviate from the average score by more than a certain threshold but not more than $\mathcal{K}$ percent of the overall operation. In this way, Transformer space shrinking can be achieved in supernet training. As a result, the following search process can be carried out more efficiently on this shrunk space.

### 3.4 Spatio-temporal Prototype Alignment

Given a query video $x_q$ and a support video $x_s$ and their embedded features $\boldsymbol{f}(x_q), \boldsymbol{f}(x_s) \in \mathbb{R}^{\mathcal{T}\times\mathcal{C}_o}$, a commonly used similarity measurement method is

$$\boldsymbol{d}\left(x_q,x_s\right) = \frac{1}{\mathcal{T}\mathcal{C}_o}\sum\left\|\boldsymbol{f}\left(x_q\right)-\boldsymbol{f}\left(x_s\right)\right\|_2^2 \tag{8}$$

where $\boldsymbol{d}(x_q,x_s)$ is the similarity distance between video $x_q$ and $x_s$, $\mathcal{T}$ and $\mathcal{C}_o$ are the temporal length and feature dimension, respectively. The feature spacing between two videos is the mean value of the difference in temporal and channel dimensions. However, this direct measurement has disadvantages in few-shot video action recognition: it is complicated to match changeable actions performed at various speeds directly and occurred at different time points in extremely few videos. To this end, we propose a nonparametric spatio-temporal feature prototype alignment method to better deal with feature matching in few-shot video action recognition. Specifically,

$$\mathcal{M}\left(x_q,x_s\right) = \text{softmax}\left[\boldsymbol{f}\left(x_q\right)\boldsymbol{f}^T\left(x_s\right)\right] \tag{9}$$

where $\mathcal{M}(x_q,x_s)$ denotes a frame-level attention map with a dimension of $T\times T$. This provides the possibility for us to calculate the query-specific prototype w.r.t the query $x_q$, such as

$$\tilde{\boldsymbol{f}}\left(x_s\right) = \mathcal{M}\left(x_q,x_s\right)\boldsymbol{f}\left(x_s\right) \tag{10}$$

where $\tilde{\boldsymbol{f}}(x_s)$ is a temporal aligned support sample representation, as shown in Figure 3. The updated feature distance between the aligned support feature and query feature can be calculated as

$$\tilde{\boldsymbol{d}}\left(x_q,x_s\right) = \frac{1}{\mathcal{T}\mathcal{C}_o}\sum\left\|\boldsymbol{f}\left(x_q\right)-\tilde{\boldsymbol{f}}\left(x_s\right)\right\|_2^2 \tag{11}$$

Then, this measurement method can be easily applied to the support set $\mathbb{S}^c$ of class $c$. i.e.

$$\tilde{\boldsymbol{d}}\left(x_q,\mathbb{S}^c\right) = \frac{1}{|\Pi|}\sum_{p\in\Pi}\tilde{\boldsymbol{d}}\left(x_q,x_p\right) \tag{12}$$

Table 1: Macro Transformer space of our SST model. Given an input video with dimension $T \times C \times W \times H$, we search over operations and the number of heads at 3 stage. Within each stage, "Choice Block" indicates the search from blocks of "TAB" and "SAB"."Spatial Downsample" represents the spatial downsampling block (SDB).

| Stage | # Layers | Operations | # Heads | Output Size |
|---|---|---|---|---|
| Patch Embedding | 4 | Convolution | - | $[T, C_1, W/16, H/16]$ |
| Choice Block | 8 | $\{TAB, SAB\}$ | 6, 12, 16 | $[T, C_1, W/16, H/16]$ |
| Spatial Downsample | 1 | $SDB$ | 12 | $[T, C_2, W/32, H/32]$ |
| Choice Block | 8 | $\{TAB, SAB\}$ | 6, 12, 16 | $[T, C_2, W/32, H/32]$ |
| Spatial Downsample | 1 | $SDB$ | 12 | $[T, C_3, W/64, H/64]$ |
| Choice Block | 8 | $\{TAB, SAB\}$ | 6, 12, 16 | $[T, C_3, W/64, H/64]$ |
| Output | 1 | Norm | - | $[T, C_o]$ |

where $x_p$ denotes the support example in support set $\mathbb{S}^c$. During training or inference, the query video $x_q$ is assigned the class of the closest query-specific prototype. This is an efficient nonparametric spatio-temporal prototype alignment method that can easily be applied to two spatio-temporal feature similarity measurements. Generating an aligned query-specific prototype for every support video can obtain more reliable prediction results in the few-shot video action classification.

### 3.5 Transformer Space Design

To find the optimal combination layout for SAB and TAB, we construct a transformer space for the FS video action classification. Let $T \times C \times W \times H$ be the dimension of the input frames of our supernet. We extract multi-scale features hierarchy in every stage with a scaling step of 2 from the input frames. These intermediate embeddings have a spatial stride of $16, 32, 64$ w.r.t the input. We do not reduce the time dimension to generate a feature representation with the same length as the input sequence. For each layer in our SST model, we search over the operation type and the number of heads in SAB and TAB. The model can not only choose what kind of attention to do in different layers, but also obtain the most appropriate number of self-attention heads.

Diagrams of our TAB and SAB are illustrated in Figure 2. The input dimensions are both $Twh \times c$, where the $w$,$h$,$c$ denote the width, height, and channel of intermediate embeddings. In the time attention block, a temporal self-attention mask with dimension $wh \times T \times T$ is computed to capture temporal dependencies across frames. Similarly, spatial attention with dimension $T \times wh \times wh$ can be obtained in the space attention block. Thus, these two modules have different functions in the process of video understanding. To facilitate the architecture search, these two modules share the same input and output dimensions. Table 1 provides a basic overview of the Transformer space.

## 4 Experiments

### 4.1 Datasets and experimental setups

**Datasets.** Our proposed method is evaluated on two popular benchmarks: UCF101 [31] and HMDB51 [21], using split from [27]. UCF101 [31] is a human action videos dataset from Youtube, which contains 13320 video clips and 101 action categories. We used the split with 70/10/21 action categories in training/validation/testing. HMDB51 [21] contains 6849 video clips divided into 51 classes. A split with 31/10/10 classes is used for HMDB51 [21]. The standard 5-way 1-shot and 5-way 5-shot is applied to evaluate the performance of our method.

**Implementation details.** For each $n$-way $k$-shot episodic task, we randomly sample $n$ class with each class containing $k$ examples as the support set. And we randomly sample one sample for each class in $n$ classes as the query set. Thus each episodic task contains $n \times (k+1)$ samples. For video preprocessing, the sparse sampling strategy is used to fetch $T$ frames for each video. During training, we resize sampled frames to $256 \times 256$ and then randomly crop a $224 \times 224$ region as input. During testing, random crop is replaced by center crop.

For supernet, we construct the base Transformer space according to Table 1. The stage and block setting (the numbers of MHSA and FC layers) is inspired by [2, 16]. Follow LeViT [16], the depth of each stage is set to the same. And the head number $h$ in the SAB and TAB is chosen within the

Table 2: Few-shot action classification results on HMDB51 [21].

| Method | Frames | 5-way 1-shot | | | 5-way 5-shot | | |
|---|---|---|---|---|---|---|---|
| | | Acc | Params | FLOPs | Acc | Params | FLOPs |
| TimeSformer [2] | 8 | 33.2 | 40.7M | 73.35G | 41.7 | 40.7M | 73.35G |
| TRX [27] | 8 | 29.1 | 25.6M | 41.43G | 46.4 | 25.6M | 41.43G |
| ARN [47] | 20 | 45.2 | - | - | 60.6 | - | - |
| | 4 | 39.2 | 8.54M | 6.83G | 57.1 | 8.53M | 6.81G |
| Ours | 8 | 51.1 | 8.89M | 13.64G | 60.4 | 8.91M | 13.65G |
| | 12 | 52.4 | 8.87M | 20.49G | 62.2 | 8.86M | 20.48G |

Table 3: Few-shot action classification results on UCF101 [31].

| Method | Frames | 5-way 1-shot | | | 5-way 5-shot | | |
|---|---|---|---|---|---|---|---|
| | | Acc | Params | FLOPs | Acc | Params | FLOPs |
| TimeSformer [2] | 8 | 42.0 | 40.7M | 73.35G | 63.0 | 40.7M | 73.35G |
| TRX [27] | 8 | 46.7 | 25.6M | 41.43G | 67.0 | 25.6M | 41.43G |
| | 4 | 60.1 | 8.61M | 6.79G | 68.2 | 8.63M | 6.83G |
| Ours | 8 | 63.8 | 8.87M | 13.72G | 69.7 | 8.84M | 13.67G |
| | 12 | 65.4 | 8.76M | 20.34G | 70.4 | 8.87M | 20.45G |

set $\{6, 12, 16\}$ to encourage each block to select its optimal head numbers freely. During training, the AdamW optimization method is used to train the supernet from scratch. All experiments are implemented with 8 Nvidia 1080Ti GPUs.

Furthermore, it should be mentioned that existing FS action classification all tend to rely on the pre-trained weights (e.g., pre-trained on ImageNet) to initialize the model. However, this pre-training may violate the basic assumption of few-shot learning that the query classes cannot be seen during meta-training [55]. We find that ImageNet contains very high-related classes to those query classes during meta-testing, such as the "Guitar" class in ImageNet vs. "PlayingGuitar" class in UCF101. This makes ImageNet pre-training unreasonable and problematic. Thus, all model optimization in this paper is based on randomly initialized weights to ensure the fairness of the comparison results.

## 4.2 Comparison with state-of-the-arts

We compare our method with existing few-shot action classification methods such as ARN [47], TRX [27] and existing video understanding methods such as TimeSformer [2]. Table 2 shows the comparison of our method to state-of-the-art in 5-way 1-shot and 5-way 5-shot problems on HMDB51. As in Table 2, our searched models outperforms ARN [47], TRX [27] and TimeSformer [2] by a large margin on the two protocols. Meanwhile, SST models have fewer parameters and FLOPs compared with previous methods. It is well known that processing more dense frames often means greater computing budgets. However, even if our SST processes dense 12 frames, it also has a good advantage over FLOPs. From these comparison results, we can also see that the results of 5-shot setting are generally higher than that of 1-shot. This is because more available video examples can help the model match the action pattern more accurately in the inference process.

The results on UCF101 dataset are listed in Table 3. On UCF101 dataset, our method also outperforms the previous methods by a large margin. In particular, when the number of input frames is 8, SST improves TRX by 2.7% in 5-shot case and by surprisingly 17.1% in 1-shot case. Note that the performance of SST with 4 input frames can exceed that of TRX with 8 frames, which demonstrates the superiority of our method in dealing with extremely few videos.

## 4.3 Ablation Study

Here, we systematically analyze the impact of different modules in our method. Specifically, we first show the effectiveness of each component under the 8 input frames and 5-way 5-shot setting in Table 6. We follow the TimeSformer[2] and LeVit [16] to construct a baseline model, in which the intermediate modules are set manually in a way similar to TimeSformer[2]. These results show that the NAS can improve the baseline method with an improvement of 2.07%, which indicates the searched model has a stronger learning ability of spatio-temporal features. The proposed Transformer Space Shrinking and Temporal Alignment contribute to an improvement of 1.37% and 2.56%, respectively. All components collaborate and complement each other, reaching 60.42% finally. In what follows, we conduct several ablation experiments to evaluate the effectiveness of each module on HMDB51 carefully.

Table 4: Compare SST with non-searched models.

| Method | Frames | 5-way 5-shot | | |
|--------|--------|-----|--------|-------|
| | | Acc | Params | FLOPs |
| Plain model | 4 | 48.5 | 8.96M | 6.94G |
| | 8 | 54.6 | 8.96M | 13.89G |
| | 12 | 56.3 | 8.96M | 20.82G |
| Ours | 4 | 57.1 | 8.53M | 6.81G |
| | 8 | 60.4 | 8.83M | 13.65G |
| | 12 | 62.2 | 8.86M | 20.48G |

Table 5: Performance evaluation of TA.

| Method | Frames | HMDB51 | |
|--------|--------|--------------|--------------|
| | | 5-way 1-shot | 5-way 5-shot |
| w/o TA | 4 | 36.7 | 52.5 |
| | 8 | 40.2 | 58.0 |
| | 12 | 46.6 | 58.9 |
| Ours | 4 | 39.2 | 57.1 |
| | 8 | 51.1 | 60.4 |
| | 12 | 52.4 | 62.2 |

Table 6: Ablations of different modules of our method on HMDB51 [21]. And we report the mean and std over 5 runs for SST.

| NAS | Transformer Space Shrinking | Temporal Alignment | 5-way 5-shot Acc |
|-----|-----------------------------|--------------------|------------------|
| | | | $54.58 \pm 0.07$ |
| ✓ | | | $56.65 \pm 0.08$ |
| ✓ | ✓ | | $58.02 \pm 0.11$ |
| ✓ | | ✓ | $59.21 \pm 0.06$ |
| ✓ | ✓ | ✓ | $60.42 \pm 0.09$ |

Figure 4: The test accuracy of supernet.

Figure 5: The test loss of supernet.

### 4.3.1 The impact of NAS

To study the impact of NAS, we conduct several experiments on hand-engineered and searched architectures. In "Plain model", we follow the TimeSformer[2] and LeVit [16] to construct a plain model, which has a structure similar to our Transformer space, but the intermediate modules are set manually. According to Figure 1, we manually set the alternating TAB and SAB modules in each stage to build the final model. Then, the same hyperparameters are used to retrain these hand-engineered and searched models. As shown in Table 4, the searched architectures surpass the non-searched version by a considerable gap while using fewer parameters and FLOPs. These results demonstrate the effectiveness of exploiting Transformer architecture in our method.

### 4.3.2 The impact of Transformer Space Shrinking

To better study the impact of Transformer space shrinking, we conduct two comparative experiments on HMDB51. We train a supernet on 1-shot task with 4 input frames. The average metrics of randomly sampled subnetworks are regarded as the metrics of supernet. The top-1 accuracy and loss on test sets are reported in Figure 4 and Figure 5, respectively. From figures, we can observe that the introduction of the Transformer space shrinking strategy not only speeds up the convergence of the supernet, but also greatly improves the performance. Theoretically, the search space is wider, search time is longer, and training speed is slower, the stronger the need for space shrinking.

### 4.3.3 The impact of Temporal Alignment (TA)

Here we show the effectiveness of the proposed Spatio-temporal Temporal Alignment strategy. For "w/o Temporal Alignment", we directly measure the distance between temporal features with the dimension of $T \times C_o$ in the method of Eq. (8). For our complete method, support video features are first aligned to generated query-specific prototype and then compared with query features. For our complete method, support video features are first aligned to generate a query-specific prototype and then compared with query features. From Table 5, we find the spatio-temporal prototype alignment improves the baseline on various settings, which indicates the alignment strategy helps the FS video

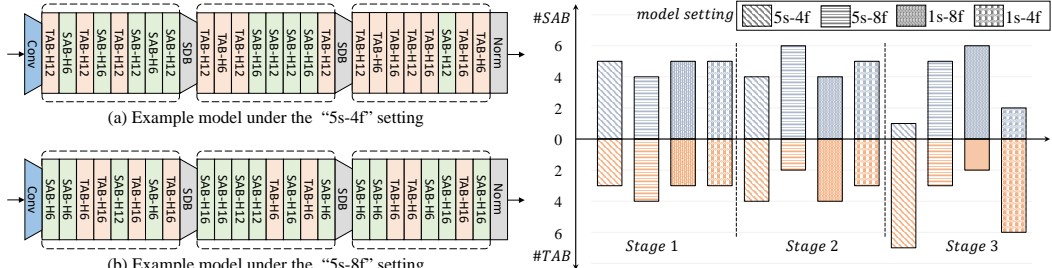

Figure 6: Visualization of searched architectures.

Figure 7: Statistics under four settings.

Table 7: Comparison results using pretrained weights on HMDB51 [21].

| Method | HMDB51 | | |
| --- | --- | --- | --- |
| | ImageNet-pretrain | UCF101-pretrain | Non-pretrain |
| TimeSformer [2] | - | 48.3 | 41.7 |
| TRX [27] | 75.6 | 54.7 | 46.4 |
| Ours | - | 67.2 | 60.4 |

action similarity measurement. For example, when using 12 frames as input, our method outperforms the non-aligned version by 5.8% on 5-way 1-shot setting. Furthermore, the proposed alignment operation is very efficient and does not introduce too much computational cost.

### 4.3.4 The impact of Pre-training

To prove the effectiveness of our method using pretrained weight, we also pretrain model on UCF101 dataset and then finetune in a few-shot learning way on HMDB51 dataset. The results are listed in Table 7. All the methods are finetuned on 5-way 5-shot tasks with 8 input frames. As shown in table, the performance of three methods can be greatly improved after pretraining on UCF101 dataset. Among them, our method still performs better than the previous two methods. In addition, we also provide the results of TRX using Imagenet pretrained weight. It can be seen from the results that the performance of TRX can also be greatly improved after pretraining with large-scale ImageNet dataset. However, as stated in section 4.1, there is a possibility that relevant information may be leaked to the unseen query class when few-shot learning methods use pretrained weights. By pretrained with a larger dataset, there may be more severe class information leakage. In this work, whether using UCF pretrained or randomly initialized weights, our method significantly outperforms the previous methods. The experiment in this subsection demonstrates that our proposed method is still better than the previous methods after pretraining to ensure that the model has better feature extraction ability.

### 4.3.5 Visualization

In addition to ablation experiments, we visualize some searched architecture and statistics under different settings. In Figure 6, two searched structures are shown in detail, in which "5s-4f" denotes the searched model under 4 input frames and 5-shot setting, "TAB-H12" denotes a TAB block with 12 self-attention heads, and so on for the other abbreviations. And Figure 7 shows some statistical results, in which the horizontal axis represents the three stages of four models, and the vertical axis represents the number of spatial and temporal blocks (positive to the SAB, negative to the TAB). It can be seen from the results that although there are some differences in the layout of TAB and SAB modules in the first two stages, the proportion of these two modules is relatively balanced. Surprisingly, we notice that the layout of the third stage is relatively large, which is quite different from the design in current hand-crafted models. Obviously, our method can search for the appropriate and specific architectures for different input dimensions and different tasks. For example, for the same 5-shot task, the models select 7 TAB modules in the third stage when 4 frames are fed in. But when 8 frames are used as input, the models only select 3 TAB modules in the third stage. We argue that the demands for the temporal modeling ability of the model are related to the time dimension sampling rate of the input data. When the model can obtain only fewer input frames, more temporal attention blocks will be placed in the third stage to improve the action modeling capability.

# 5 Conclusion

This paper introduced a FS action recognition method for a neural architecture search method with a Transformer space shrinking strategy and spatio-temporal prototype alignment. The discovered architectures can learn steadier frame-level representations from FS video samples. The spatio-temporal features are then utilized in prototype alignment operation for learning similarity between query and support actions. The query-specific class prototypes for each query sequence are generated to deal with various actions at different speeds and temporal offsets. An extensive set of ablations shows how optimal video Transformer architectures are searched, the benefits of Transformer space shrinking, and the importance of the spatio-temporal prototype alignment. In future studies, we will delve into the relationship between pretraining and few-shot learning. Avoiding the possible leakage of category information while using pretrained weights is still a matter of research.

## Acknowledgement

This work was supported in part by the Australian Research Council under Project DP210101859 and the University of Sydney Research Accelerator (SOAR) Prize.

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
