# Supplementary Materials of
# *Searching for Better Spatio-temporal Alignment in Few-Shot Action Recognition*

**Yichao Cao**[1*]**, Xiu Su**[2*]**, Qingfei Tang**[3]**, Shan You**[4]**, Xiaobo Lu**[1]**, Chang Xu**[2†]
[1]School of Automation, Southeast University,
[2]School of Computer Science, Faculty of Engineering, The University of Sydney,
[3]Enbo Technology Co.,Ltd., China,
[4]SenseTime Research
caoyichao@seu.edu.cn, xisu5992@uni.sydney.edu.au, qingfeitang@gmail.com
youshan@sensetime.com, xblu@seu.edu.cn, c.xu@sydney.edu.au

The supplementary materials are organized as follows. In Appendix A, we provide the implementation details of training and inference settings. We illustrate the details of Transformer space shrinking in Appendix B. The details of the evolutionary search are presented in Appendix C. We report the effect of Transformer space shrinking strategy with bigger Transformer space in Appendix D. Then, we implement the Patch Embedding layer in our SST with single-layer big kernel convolution layer and evaluate the performance in Appendix E. Moreover, we compare searched models with four hand-designed architectures in Appendix F. In Appendix G, we conduct the visualization for spatio-temporal feature distribution. We show more detailed searching results in Appendix H. We elaborate the relationship between spatio-temporal representation and temporal alignment in Appendix I. More results under the condition of similar compute budget with TRX are reported in Appendix J. And we also present motivations and more details of search space in Appendix K and Appendix L.

## A   Details of Training and Inference

In this section, we present the training details of our SST w.r.t. different datasets. We conduct experiments on two benchmark datasets: UCF101 [33] and HMDB51 [23]. UCF101 contains about 9154 training videos and 2745 test videos from 101 classes. HMDB51 contains about 4280 training videos and 1292 test videos from 51 classes. For the video preprocessing, we randomly sample $4/8/12$ consecutive frames to represent the input video. We follow the TRX [29] to carry out data augmentation. The input video is firstly resized to $256 \times 256$, and then randomly cropped to $224 \times 224$ pixels.

We build the Transformer space of supernet layer by layer according to Table 1. The core of this framework is the optional temporal and spatial attention module. We hope to explore the better Transformer architecture for few-shot video action recognition through architecture search. With the input layer of model, four convolutional layers are used as patch embedding layers, with kernel size of $3 \times 3$, and stride of 2. In Appendix-E, we also compare our approach with the method using single-layer large kernel convolution with a kernel size of $16 \times 16$. The whole supernet is divided into three stages by two SDB modules. Inspired by TimeSformer[2] and LeVit [17], we set up 8 layers in each stage. For each SDB module, the model will conduct spatial attention downsampling to reduce the height and width of the feature dimension by half. Finally, the model will output spatio-temporal representations as long as the input sequence for subsequent matching and alignment. At the last layer of model, Layer Normalization is used to speed up the training of network. After constructing the Transformer space for SST, we fully train the supernet and evaluate the performance of subnets on the test set.

---

[*]Equal contribution.
[†]Corresponding author.
36th Conference on Neural Information Processing Systems (NeurIPS 2022).

Table 8: Bigger Transformer space of our SST model. Residual Block (RB) is added as a new Choice Block. "#Dim" represents the intermediate MLP dimension in RB.

| Stage | # Layers | Operations | # Heads/Dims | Output Size |
|---|---|---|---|---|
| Patch Embedding | 4 | Convolution | - | $[T, C_1, W/16, H/16]$ |
| Choice Block | 8 | $\{TAB, SAB, RB\}$ | $\{6, 12, 16\}/\{256, 512, 1024\}$ | $[T, C_1, W/16, H/16]$ |
| Spatial Downsample | 1 | $SDB$ | 12 | $[T, C_2, W/32, H/32]$ |
| Choice Block | 8 | $\{TAB, SAB, RB\}$ | $\{6, 12, 16\}/\{256, 512, 1024\}$ | $[T, C_2, W/32, H/32]$ |
| Spatial Downsample | 1 | $SDB$ | 12 | $[T, C_3, W/64, H/64]$ |
| Choice Block | 8 | $\{TAB, SAB, RB\}$ | $\{6, 12, 16\}/\{256, 512, 1024\}$ | $[T, C_3, W/64, H/64]$ |
| Output | 1 | Norm | - | $[T, C_o]$ |

During training, we follow the one-shot NAS paradigm [18] to optimize the supernet by uniform path sampling. After the data is fed into the model, one block will be randomly selected as the operation of each layer. In this way, all subnets and their weights are trained fully and equally. In our setup, the training process uses a total of 20 epochs. The training set is used to optimize the weight $w_a$ of the subnet $a$. Firstly, we let the network warm up for 3 epochs so that the each block in supernet has an initial weight. In the subsequent epochs, uniform path sampling and Transformer space shrinking are jointly utilized to facilitate the training of the supernet. During inference, we follow TRX [29] to perform episodic tasks for few-shot action recognition, in which there are 1 query and $1/5$ support videos for each class. In each batch, we randomly sample five categories for inference. $N$ input videos with length $T$ will be transformed into $N$ spatio-temporal features with the same length $T$ through the subnet. The training takes 1 day on 8 Nvidia 1080Ti GPUs with our PyTorch implementation.

# B  Details of Transformer Space Shrinking

To facilitate the training and searching process for supernet, we introduce the Transformer space shrinking strategy. There are 24 searchable layers in our SST model, and 6 choice blocks in each layer. Thus, we need to select 24 optimal operations from 144 choice blocks. We follow the Eq. (6) and Eq. (7) to fairly evaluate all the 144 operations. During this process, the effects on accuracy and FLOPs of the whole subnet are both taken into account. After the warming-up in training process, we conduct the shrinking operations every two epochs. In this work, we set the shrink percentage and score threshold as 10% and 0.2, respectively. We conduct an operation mask for all operations to control the selectivity of supernet to operations. The initial value of the mask is set to 1, indicating that the operation can be selected. During the training, the loss value and structure of each subnet will be recorded. We calculate the score of all candidate operations for every two epochs according to Eq. 6. Then, all operations are ranked according to their scores, and the top 10% of operations are discarded. The mask values corresponding to the discarded operations will be set to 0, indicating that these operations are in an unselectable state in the subsequent search. In the initial stage of shrinking, as there are more low-performance operations in Transformer space, so more operations will be discarded during this period. With the increase of shrinking times, the whole space will gradually evolve to a more compact state, and the operation discarding speed will gradually slow down. In this way, the average performance of operations reserved in the shrunk space becomes higher, which speeds up the convergence speed and performance of supernet and also speeds up the search speed of subsequent evolutionary search.

# C  Details of Evolutionary Search

To avoid the heavy search from the enormous Transformer space e.g. $4.74 \times 10^{18}$ for our SST, the multi-objective NSGA-II algorithm is adopted to implement the search. In detail, the population size and the maximum iteration are set as 10 and 50, respectively. To carry out an architecture search, the initial population of 10 individuals is randomly generated from shrunk Transformer space. Then, we evaluate the initial population on the test set and use the tournament selection algorithm to select the 10 subnet codes reserved for each generation. Two-point crossover and polynomial mutation are used to generate the population for the next iteration. Finally, we select the subnet architecture with the best performance in the whole search process and then retrain it from scratch.

Table 9: Performance evaluation of proposed space shrinking strategy on a larger Transformer space.

| Space Shringking | Larger Space | 5-way 1-shot | | | 5-way 5-shot | | |
|---|---|---|---|---|---|---|---|
| | | Acc | Params | FLOPs | Acc | Params | FLOPs |
| | ✓ | 45.7 | 8.47M | 12.82G | 56.3 | 8.55M | 12.78G |
| ✓ | ✓ | 49.3 | 8.52M | 12.84G | 58.1 | 8.67M | 12.96G |
| ✓ | | 51.1 | 8.89M | 13.64G | 60.4 | 8.91M | 13.65G |

Table 10: Comparison results of two Patch Embedding approaches.

| Method | Frames | HMDB51 | |
|---|---|---|---|
| | | 5way-1shot | 5way-5shot |
| single layer | 8 | 50.7 | 60.1 |
| 4-layer | 8 | 51.1 | 60.4 |

## D  Effect of Transformer Space Shrinking with Bigger Transformer Space

As described in Section 4.3.2, the larger the Transformer space, the greater the difficulty of model training and search in theory. Thus, we also evaluate the performance of the proposed shrinking strategy in bigger Transformer space. First, we build a larger space, in which Residual Block (RB) is added besides TAB and SAB as shown in Table 8. In RB module, we do not carry out spatio-temporal attention operation but only design a simple residual operation. We allow the model to independently select the intermediate dimension of MLP in RB from the set $\{256, 512, 1024\}$. In this way, there are more choices to construct neural networks in each stage. Typically, with the introduction of three RB operations, the transformer space of operations is increased from $4.74 \times 10^{18}$ to $7.98 \times 10^{22}$. After building the Transformer space, we get the optimal model according to the training and search process described above, and then retrain the optimal model for evaluation. The comparisons on HMDB51 [23] are shown in the Table 9. It is observed that after using the shrinking strategy in a larger space, the accuracy of searched model is improved by 3.6% and 1.8% respectively in 5-way 1-shot and 5-way 5-shot tasks. This is because the shrinking operation can effectively reduce the probability of selecting low-quality operations, reduce the Transformer space size and retain high-quality operations, to improve the search efficiency. Furthermore, by comparing the latter two groups of experiments, it is found that the accuracy on a large Transformer space is about 2 percentage points lower than that on a small Transformer space. These results suggest that a larger space is not necessarily better, and rational design is needed to ensure the final searched performance. And these comparisons also demonstrate the superiority of the proposed space in section 3.5.

## E  Implementing the Patch Embedding with Single Convolution Layer

In our SST model, we refer to the LeViT's [17] design and adopt 4-layer CNN as patch embedding layer. However, in other vision Transformer architectures, patch embedding layer is also be achieved through a large kernel convolutional layer. To explore the effect of the two methods, we also set a contrast experiment on HMDB51 [23] as shown in Table 10. The input and output dimensions of a single-layer large kernel convolution with a kernel size of $16 \times 16$, and stride of 16 can be the same as that of 4-layer CNN a kernel size of $3 \times 3$, and stride of 2. From the experimental results, the 4-layer CNN performs slightly better than the single layer method. Thus, we adopt the former in the final structure design.

## F  Comparisons to Hand-designed Architectures

In this section, we also provide comparison results with four hand-designed architectures. Here, four hand-designed architectures, which are denoted as Model-1, Model-2, Model-3 (Figure 9-a) and Model-4 (Figure 9-b). Among them, LeViT [17] is adopted as the backbone in Model-1 to extract spatial features from videos, and then temporal alignment and matching are implemented for FS action recognition. Model-2 is a typical TimeSformer architecture that consists of 12 Divided Space-Time Attention (DSTA) blocks. The difference between Model-1 and Model-2 is that the former can only extract spatial features, while the latter can take into account temporal information. Model-3 is a TimeSformer architecture equipped with Space Downsampling Blocks (SDB) between different stages, which also captures the spatio-temporal feature via DSTA block. The introduction of SDB is to

Table 11: Comparisons to four hand-designed architectures on HMDB51.

| Method | Backbone | HMDB51 | |
| --- | --- | --- | --- |
| | | 5way-1shot | 5way-5shot |
| Model-1 | LeViT | 28.6 | 43.2 |
| Model-2 | TimeSformer | 33.2 | 41.7 |
| Model-3 | TimeSformer-SDB | 34.7 | 43.2 |
| Model-4 | Plain model | 43.8 | 54.6 |
| Ours | Searched model | 51.1 | 60.4 |

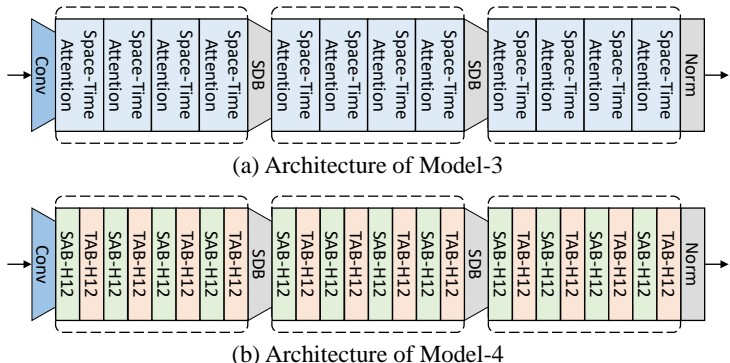

(a) Architecture of Model-3

(b) Architecture of Model-4

Figure 8: Visualization of two hand-designed architectures.

reduce the intermediate embedding dimensions and thus reduce the FLOPs of model. Model-4 is same as the plain model in Section 4.3.1, in which alternating TAB and SAB are manually set to capture spatio-temporal features. The comparison results are reported in Table 11. From these comparison results, we can conclude that the time attention is very important for the video recognition tasks. Moreover, few-shot video recognition places a higher demand on video understanding architecture design, and hand-designed architectures are hardly superior to searched models.

# G    Visualization of Spatio-temporal Feature Distribution

The capacity of spatio-temporal feature representation is crucial for few-shot action recognition, as it determines the coherence of the temporal patterns. A stronger feature extraction ability is helpful for the model to learn distinctive feature representation from extremely few samples. To verify the performance of our searched model and baseline methods, t-SNE method is adopted to visualize the feature distribution. We randomly select five categories from the testing set of UCF101 [33], each category contains 1000 videos. The feature dimensions of TimeSformer [2], TRX [29] and our methods are 768, 2048 and 384 respectively. As shown in Figure 9, compared with TimeSformer [2] and TRX [29], the feature distribution of our method has better intra-class compactness and inter-class dispersion, which indicates that our method has better feature representation ability.

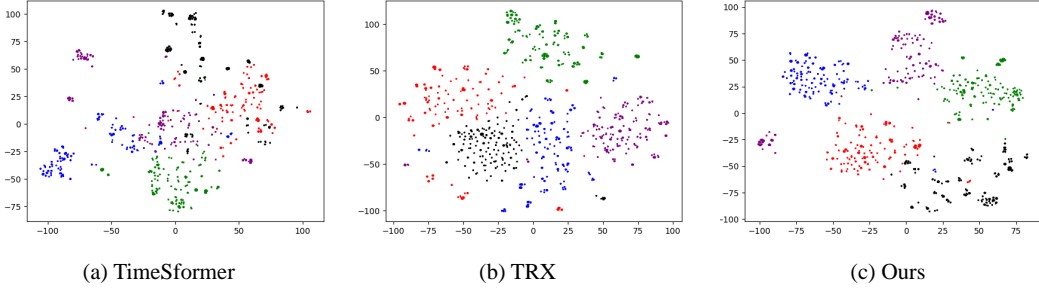

(a) TimeSformer                    (b) TRX                    (c) Ours

Figure 9: t-SNE visualization of feature distribution on UCF101 [33] testing set. Different colors represent different classes.

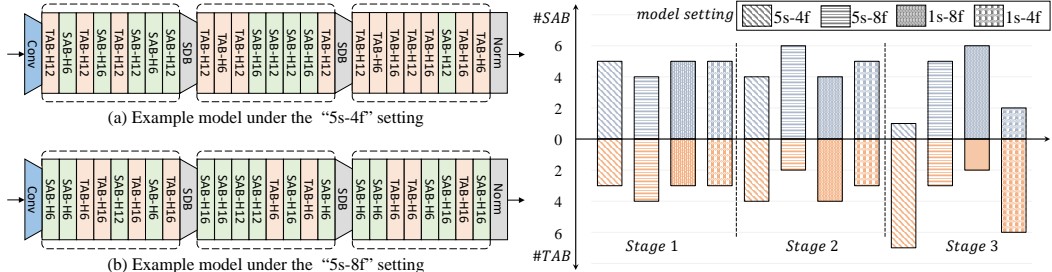

Figure 10: Visualization of searched architectures.   Figure 11: Statistics under four settings.

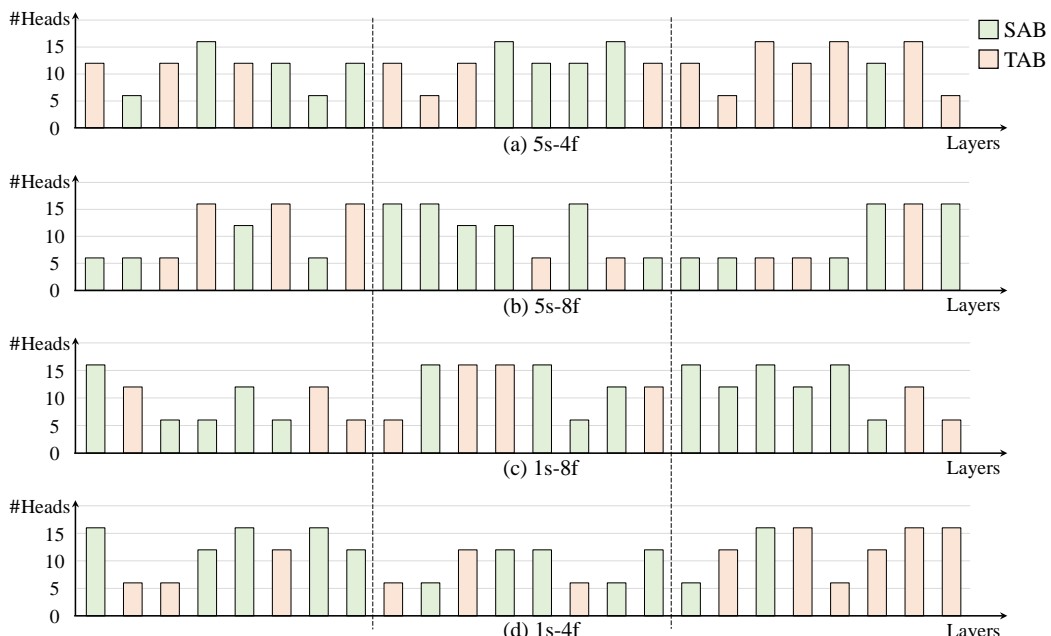

Figure 12: More visualization of searched networks w.r.t. different settings. The vertical axis means the number of self-attention heads in TAB and SAB modules.

# H   More Visualization of Searched Architectures

We visualize some searched architecture and statistics under different settings. In Figure 10, two searched structures are shown in detail, in which "5s-4f" denotes the searched model under 4 input frames and 5-shot setting, "TAB-H12" denotes a TAB block with 12 self-attention heads, and so on for the other abbreviations. And Figure 11 shows some statistical results, in which the horizontal axis represents the three stages of four models, and the vertical axis represents the number of spatial and temporal blocks (positive to the SAB, negative to the TAB). It can be seen from the results that although there are some differences in the layout of TAB and SAB modules in the first two stages, the proportion of these two modules is relatively balanced. Surprisingly, we notice that the layout variance of the third stage is relatively large, which is quite different from the design in current hand-crafted models. We argue that the demands for the temporal modeling ability of the model are related to the time dimension sampling rate of the input data. When fewer input frames are fed in, more temporal attention blocks will be placed in the third stage to improve the action modeling capability.

For intuitively understanding, the number of heads in TAB and SAB in our searched networks are shown in Figure 12. Note that the optimal model is different when under different input conditions and different few-shot task settings. This indicates that different tasks require different spatio-temporal modeling capabilities. Our proposed method can ensure that the model architecture can be flexibly organized and adjusted in a given search space to solve diverse challenges. And such a flexible and detailed structural design can not be achieved by manual design.

Table 12: Experimental results without pre-trained weights at similar compute budget.

| Method | UCF101 (5-way 5-shot) | | |
| --- | --- | --- | --- |
| | Acc | Params | FLOPs |
| TimeSformer [2] | 63.0 | 40.7M | 73.35G |
| TRX [25] | 67.0 | 25.6M | 41.43G |
| Ours | 69.7 | 8.84M | 13.76G |
| Ours (Searchable layers×3) | 70.2 | 26.1M | 41.19G |

# I Relationship between Spatio-temporal Representation and Temporal Alignment

FS action recognition aims at the recognition of action categories in extremely few videos. This requires the model to learn the latent action information from a few samples (such as one video). However, human actions are very complex and diverse in time and space dimensions. An action often contains many sub-actions, such as a high jump consisting of running and jumping. Therefore, the primary premise of FS action recognition is to learn a set of spatio-temporal representations that can reliably reflect the human body's spatial action and motion. In addition, different video action categories and parameter settings will affect the task difficulty and model complexity. The simultaneous influence of many external factors such as action duration, frame rate, and so on will make the process of manually designing architectures very cumbersome, and the hand-designed models are difficult to surpass the searched models. Therefore, we propose to pay attention to spatiotemporal representations and achieve more reliable spatiotemporal feature extraction through architecture search. On the other hand, the problem of human action matching is also complex. Since the start, stop time, and duration of actions in the video are not fixed, and the time points of sub actions are also changeable. To this end, how to efficiently and reliably align and match video features of the same category is the second focus of this work. In summary, spatio-temporal representation and temporal alignment jointly influence the performance of FS action recognition tasks, and both are necessary.

# J More Results with Similar Budget

Moreover, we compared the performance of our method and the previous methods under the condition of similar compute budget on UCF101 dataset. We increase the number of searchable layers of our model to three times of the original version stage by stage, making its Params and FLOPs similar to that of TRX. Then we evaluate the performance of our model in 5-way 5-shot setting. All methods here use 8 frames as input. As shown in Table 12, our method still surpasses TRX and TimeSformer and is even slightly better than the original version. This verifies that our method is still better than the previous method at same budget.

# K Motivations for Our Search Space

In the process of designing search space, we drew from many prior excellent works. First, in terms of the overall structure of the model, [14] proposes that the video understanding model has different emphasis on the resolution of timing and features in different stages. And the proposed X3D model manually designed has achieved great success in video understanding task. Few-shot action recognition places high demands on the ability of the video representation, which motivated us to utilize NAS to explore the model structure. We hope our model can spontaneously choose and focus on different types of information at different stages to obtain better representations. Second, in terms of video understanding through NAS, [22] and [54] explored the method of searching 3DCNN, and both achieved good performance. Considering the natural advantages of Transformer in sequence analysis, we plan to design search space based on Transformer. Third, through the comparison of various space-time modules, the manually designed TimeSformer confirms the effectiveness of the Divided Space-Time Attention module. Finally, we extract independent Space "SAB" and Time Attention Blocks "TAB" to build the final search space.

## L   The Selection of #Head in Search Space

Our initial design goal is that the search space can accommodate both video understanding and feature extraction ability. And selecting the # head is mainly based on the consideration of search complexity. Since the actual performance of NAS is determined by the space complexity and search efficiency. Considering too many dimensions may affect the performance of the searched model. In some previous Transformer NAS methods (e.g. AutoFormer[6], ViTAS [38]), it is mentioned that # head and channel dimension are indeed important for model design. But comparatively speaking, the search for # head is not easy to lead to huge search space. This is one of its advantages. In this work, the size of our search space is $4.74 \times 10^{18}$. If we follow the setting of ViTAS[38] to search the token embedding dimension, the search space may grows to $4.74 \times 10^{42}$. If we follow the setting of AutoFormer[6] to search the embedding dimension and MLP ratio (the ratio of hidden dimension to the embedding dimension in the multi-layer perceptron), the search space will become larger (even $6.32 \times 10^{51}$). Therefore, we finally referred to the settings of TimeSformer[2] in the channel dimension without conducting channel search.

For spatio-temporal resolution, we divide the model into three stages with different resolutions, giving the model some freedom to choose spatio-temporal resolution. We also found that, when fewer input frames are fed in, more temporal attention blocks will be placed in the third stage to improve the action modeling capability.