# OpenReview forum: "Searching for Better Spatio-temporal Alignment in Few-Shot Action Recognition"
_NeurIPS.cc/2022/Conference — NeurIPS 2022 Accept_

### Official Review · Reviewer_4Tcp · 2022-07-11

**Rating:** 6
**Confidence:** 4
**Soundness:** 2 fair
**Presentation:** 2 fair
**Contribution:** 2 fair

**Summary:**

This paper presents a neural architecture search method for few-shot action recognition. The main contribution is the proposal of transformer space shrinking strategy and spatio-temporal prototype alignment. Experiments are conducted on HMDB51 and UCF101.

**Questions:**

1. Performance comparison with the state-of-the-art methods when using pre-trained weights
2. Experimental results on large-scale datasets

**Limitations:**

See above

**Strengths And Weaknesses:**

Strengths:
1. Ablation studies are conducted to evaluate the effectiveness of each component.
2. The proposed transformer space shrinking strategy reduces the time and cost for architecture search.
Weaknesses:
1. I understand that there may be some contradictions between pre-training and few-shot learning. However, since almost all of the state-of-the-art methods are based on pre-trained weights instead of random initialization, the authors should also report performances with pre-trained weights to ensure comparability among them.
2. To demonstrate the superiority and generality of the proposed method, experimental evaluations should be performed on large-scale datasets of action recognition, such as Kinetics and Something-Something, just like the competitors did.
3. Some closely related works, such as [a, b], should also be introduced and compared in the paper.
[a] Xiatian Zhu, Antoine Toisoul, Juan-Manuel Perez-Rua, Li Zhang, Brais Martinez, and Tao Xiang. "Few-shot action recognition with prototype-centered attentive learning." In BMVC, 2021.
[b] Shuyuan Li, Huabin Liu, Rui Qian, Yuxi Li, John See, Mengjuan Fei, Xiaoyuan Yu, and Weiyao Lin. "TA2N: Two-Stage Action Alignment Network for Few-Shot Action Recognition." In AAAI, 2022.

---

> ### Author Response · Authors · 2022-08-02
> **Responses to Reviewer #3**
>
> Thanks for your positive support and constructive opinions. The following answers will be revised accordingly in the final version. And we will discuss related papers (e.g., PAL and TA2N) in our final version.
>
> Q1: I understand that there may be some contradictions between pre-training and few-shot learning. However, since almost all of the state-of-the-art methods are based on pre-trained weights instead of random initialization, the authors should also report performances with pre-trained weights to ensure comparability among them.
>
> A1: Thanks for your valuable advice. We have supplemented relevant experiments in rebuttal.
> Because the backbone of our model synchronously learns spatio-temporal features, it is a model of video understanding type. We can only use video datasets to train it. We report the performance of our model after pretraining on large-scale Kinetics-400 dataset. Similarly, all models here use 8 frames as input. Through pretraining on Kinetics dataset, all models are compared under a fair pretraining condition. After that, the few-shot dataset UCF101 is used for fine-tuning and testing, and the results are shown in the following table. As shown in Table 12, our proposed model still has the best performance after pretraining on large-scale dataset. With only 1/3 FLOPs, our method can surpass TRX by 2.5% and 10.8% on the UCF101 and HMDB51 dataset, respectively.
>
> |     Method             |     Pretraining     |             |     UCF101    |               |             |     HMDB51    |               |
> |------------------------|---------------------|-------------|---------------|---------------|-------------|---------------|---------------|
> |                        |                     |     Acc     |     Params    |     FLOPs     |     Acc     |     Params    |     FLOPs     |
> |     TimeSformer [2]    |     -               |     63.0    |     40.7M     |     73.35G    |     41.7    |     40.7M     |     73.35G    |
> |     TimeSformer [2]    |     Kinetics-400    |     80.5    |     40.7M     |     73.35G    |     54.2    |     40.7M     |     73.35G    |
> |     TRX [25]           |     -               |     67.0    |     25.6M     |     41.43G    |     46.4    |     25.6M     |     41.43G    |
> |     TRX [25]           |     Kinetics-400    |     85.1    |     25.6M     |     41.43G    |     60.7    |     25.6M     |     41.43G    |
> |     Ours               |     -               |     69.7    |     8.84M     |     13.76G    |     60.4    |     8.91M     |     13.65G    |
> |     Ours               |     Kinetics-400    |     87.6    |     8.73M     |     13.61G    |     71.5    |     8.75M     |     13.52G    |
>
> Due to page limitations of manuscript, these experiment results are updated to the Supplementary Materials, from Line 625 to Line 631. We are very sorry for these late experiments.
>
> Q2: To demonstrate the superiority and generality of the proposed method, experimental evaluations should be performed on large-scale datasets of action recognition, such as Kinetics and Something-Something, just like the competitors did.
>
> A2:Thanks for your valuable advice. We supplemented the experiment of pre-training with large-scale Kinetics dataset, and proved the superiority of our method under the same pre-training conditions. In addition, on the two classic FSL datasets, UCF101 and HMDB51, we conducted a large number of fair experiments to prove the effectiveness of the method, including using and not using pre training weights. We also follow the suggestions of other reviewers to supplement the comparative experiment of training similar compute budget models from scratch to reduce the risk that large models are easier to overfit.
>
> Q3: Some closely related works, such as [a, b], should also be introduced and compared in the paper.
>
> A3: Thanks for your valuable advice. We are sorry for missing related references. According to your suggestion, we have revised the RELATED WORK accordingly, together with related references. Two relevant references are added into this subsection:
>
> [43]	Zhu, X., Toisoul, A., Perez-Rua, J.M., Zhang, L., Martinez, B., Xiang, T.: Few-shot action recognition with prototype-centered attentive learning. arXiv preprint arXiv:2101.08085 (2021)
>
> [23]	Li, S., Liu, H., Qian, R., Li, Y., See, J., Fei, M., Yu, X., Lin, W.: Ta2n: Two-stage action alignment network for few-shot action recognition. In: Proceedings of the AAAI Conference on Artificial Intelligence. vol. 36, pp. 1404–1411 (2022)
>
> And, we added the content “[43] proposes a prototype-centered contrastive learning loss and a hybrid attentive learning mechanism to minimize the negative impacts of outliers and promote class separation in few-shot action recognition task. [23] designs temporal transform module to handle action misalignment, which consists of two parts: a localization network and a temporal affine transformation.” to the revised manuscript, from Line 75 to Line 79.

---

> > ### Comment · Reviewer_4Tcp · 2022-08-08
> > **Change rating to weak accept**
> >
> > I appreciate the effort made by the authors to provide additional experimental results. The response addressed all of my concerns. Therefore, I raise my rating to weak accept.

---

> ### Author Response · Authors · 2022-08-06
> **Further discussion with reviewer 4Tcp**
>
> Dear reviewer 4Tcp:
>
> We thank you for the precious review time and valuable comments. We have provided corresponding responses and results, which we believe have covered your concerns. We hope to further discuss with you whether or not your concerns have been addressed. Please let us know if you still have any unclear parts of our work.
>
> Best,

---

> ### Author Response · Authors · 2022-08-08
> **Further discussion with reviewer 4Tcp**
>
> Dear reviewer 4Tcp:
>
> We sincerely thank you for your careful reviews and insightful comments. We have tried our best to respond to the questions raised and add corresponding experimental results to the updated manuscript and supplementary. Please let us know if you still have any unclear parts of our work. We will still try our best to respond and improve. Thanks again for your efforts to review our work.
>
> Best,

---

### Official Review · Reviewer_bQCA · 2022-07-12

**Rating:** 7
**Confidence:** 4
**Soundness:** 4 excellent
**Presentation:** 4 excellent
**Contribution:** 3 good

**Summary:**

This paper proposes a spatio-temporal feature matching and alignment method for the task of few-shot action recognition with the end-to-end design of models. To achieve this aim, the authors construct a transformer space with spatial and temporal attention elements and search for the optimal Spatio-Temporal representations from few-shot videos from the architecture perspective. In detail, this paper introduces a transformer space shrinking strategy to evolve transformer space and speed up the architecture search adaptively. Besides, this paper proposes a more efficient and general spatio-temporal prototype alignment method, which can be conveniently adapted to arbitrary length video matching. This paper is novel, and the promising results compared to baseline methods prove the efficientness of the proposed method.

**Questions:**

As illustrated in the main review, I curious more about the code, computation budget, and some ablation studies of this algorithm.


**Limitations:**

Yes, the authors adequately discussed the limitations of this paper.

**Strengths And Weaknesses:**

+ The formulation of transformer space shrinking is good. It is interesting and reasonable that this paper leverages the expectation of subnets and budgets for evaluating different operations. However, this method is a little complicated. Do authors have any plan to open source the code for the contribution of the community?

+ The authors propose a spatio-temporal prototype alignment method. I am considering the efficiency of this new method. Does the proposed method have the same computation budget as the old way?

+ The visualization of searched architectures is very impressive, and the search architectures' analysis seems promising.

+ With much smaller FLOPs and parameters, this algorithm achieves relatively good performance compared to the baseline methods.

- This paper only provides the experiments without pre-trained weights. I am curious about the results of the pre-trained weights with the ImageNet dataset. With this setting, can this algorithm still achieve a promising result?

- Fig 4 and Fig 5 analyze the effect of search space shrinking from the aspect of supernet. I notice that the test loss in Fig 5 starts from 2.5 epochs. Why not show the results from the 0 epoch? Besides, I do not think there is a direct correspondence between the training, testing, and removing useless operations. The supernet may get better training because of the smaller search space rather than removing redundant ops.

---

> ### Author Response · Authors · 2022-08-02
> **Responses to Reviewer #2 (Part 1)**
>
> Thanks for your positive support and constructive opinion. We will enhance our writing, and the following answers will be revised accordingly in the final version.
>
> Q1: The formulation of transformer space shrinking is good. It is interesting and reasonable that this paper leverages the expectation of subnets and budgets for evaluating different operations. However, this method is a little complicated. Do authors have any plan to open source the code for the contribution of the community?
>
> A1: We do have plans to open source. The code will be available in GitHub after camera-ready, and all experimental details and procedures will be disclosed.
>
> Q2: The authors propose a spatio-temporal prototype alignment method. I am considering the efficiency of this new method. Does the proposed method have the same computation budget as the old way?
>
> A2: Thanks for your advice. Our method is more efficient than the old way. For example, TRX aggregates the temporal information through the arrangement and combination of spatial information pairs/triplets. The complexity will increase rapidly with the number increase of the input frames. Moreover, this combination of sparse frames is not suitable for processing long videos, and its recognition ability for complex actions will also be limited (some complex human actions cannot be represented by only 2 or 3 sparse sampled frames). A simple example is that pairs/triplets-based approaches cannot distinguish whether a person hits the desk 3 times or 4 times. Because the maximum number of sampling frames is only 3, and it is difficult for the model to understand this repetitive action beyond triplet. Our method is to directly generate frame level spatio-temporal representation, which already contains rich temporal information, so the laborious combination operation is omitted. It can be considered as a more concise and general video feature alignment method. Here we make a table to show the combinatorial complexity explosion faced by the alignment method in TRX.
>
> |     # Input Frames    |     TRX [28]    |      TRX [28]    |            Ours           |
> |:---------------------:|:---------------:|:----------------:|:-------------------------:|
> |                       |      #Pairs     |     #Triplets    |     Temporal Dimension    |
> |            4          |         6       |         4        |              4            |
> |            6          |        15       |         20       |              6            |
> |            8          |        28       |         56       |              8            |
> |           12          |        66       |        220       |             12            |
> |           16          |        120      |        560       |             16            |

---

> ### Author Response · Authors · 2022-08-02
> **Responses to Reviewer #2 (Part 2)**
>
> Q3: This paper only provides the experiments without pre-trained weights. I am curious about the results of the pre-trained weights with the ImageNet dataset. With this setting, can this algorithm still achieve a promising result?
>
> A3: Thanks for your valuable advice. We have supplemented relevant experiments in rebuttal.
> Because the backbone of our model synchronously learns spatio-temporal features, it is a model of video understanding type. We can only use video datasets to train it. We report the performance of our model after pretraining on large-scale Kinetics-400 dataset. Similarly, all models here use 8 frames as input. Through pretraining on Kinetics dataset, all models are compared under a fair pretraining condition. After that, the few-shot dataset UCF101 is used for fine-tuning and testing, and the results are shown in the following table. As shown in Table 12, our proposed model still has the best performance after pretraining on large-scale dataset. With only 1/3 FLOPs, our method can surpass TRX by 2.5% and 10.8% on the UCF101 and HMDB51 dataset, respectively.
>
> |     Method             |     Pretraining     |             |     UCF101    |               |             |     HMDB51    |               |
> |------------------------|---------------------|-------------|---------------|---------------|-------------|---------------|---------------|
> |                        |                     |     Acc     |     Params    |     FLOPs     |     Acc     |     Params    |     FLOPs     |
> |     TimeSformer [2]    |     -               |     63.0    |     40.7M     |     73.35G    |     41.7    |     40.7M     |     73.35G    |
> |     TimeSformer [2]    |     Kinetics-400    |     80.5    |     40.7M     |     73.35G    |     54.2    |     40.7M     |     73.35G    |
> |     TRX [25]           |     -               |     67.0    |     25.6M     |     41.43G    |     46.4    |     25.6M     |     41.43G    |
> |     TRX [25]           |     Kinetics-400    |     85.1    |     25.6M     |     41.43G    |     60.7    |     25.6M     |     41.43G    |
> |     Ours               |     -               |     69.7    |     8.84M     |     13.76G    |     60.4    |     8.91M     |     13.65G    |
> |     Ours               |     Kinetics-400    |     87.6    |     8.73M     |     13.61G    |     71.5    |     8.75M     |     13.52G    |
>
> Due to page limitations of manuscript, these experiment results are updated to the Supplementary Materials, from Line 625 to Line 631. We are very sorry for this late experiment.
>
> Q4: Fig 4 and Fig 5 analyze the effect of search space shrinking from the aspect of supernet. I notice that the test loss in Fig 5 starts from 2.5 epochs. Why not show the results from the 0 epoch? Besides, I do not think there is a direct correspondence between the training, testing, and removing useless operations. The supernet may get better training because of the smaller search space rather than removing redundant ops.
>
> A4: Thanks for your valuable advice. This is because the loss changes dramatically in the initial stage of training. Within the initial few epochs, loss often drops several times rapidly, e.g., 8.7->1.5. However, this figure is to highlight the differences of the model after it gradually tends to be stabilized. So we omit the performance of the first two epochs to highlight the overall performance of the follow-up.

---

### Official Review · Reviewer_kc2E · 2022-07-21

**Rating:** 6
**Confidence:** 4
**Soundness:** 3 good
**Presentation:** 3 good
**Contribution:** 3 good

**Summary:**

This paper presents a video model for few-shot action recognition. It has two main contributions: (1) NAS to search for model components from a Transformer search space w/ divided space-time attention (space: selecting Spatial/Temporal attention blocks, selecting attention heads in {6,12,16}), and (2) a Prototype Alignment module to compute similarity scores between query and support sets (similar to TRX [25]). Transformer search space is reduced during training based on a shrinkage strategy. The corresponding shrinkage score has the following intuition to determine the strength of an operator (i.e., S/T attention or different number of heads): If the expected loss of a subnet with a certain operator deviates (beyond a threshold) from the expected loss of subnets with the same compute budget, then the corresponding operator is dropped from the search space. Simply put, an operator is useless if it performs worse at the same budget. The authors validate their method with few-shot settings in UCF101 and HMDB51, but mainly consider models initialized from scratch (w/o any ImageNet pretraining) in contrast to previous work. Ablation studies further provide insights into different components of the proposed design.

**Questions:**

- Why is this the best search space? This is not well-motivated. I believe the authors should discuss why these dimensions (spatial/temporal/heads) are the ones to search in and whether it is sufficient. There are many other aspects such as spatio-temporal resolution, channel-expansion, parallel/multi-path connections, components beyond self-attention. I agree that including all these axes will be practically impossible, but the choices of authors should be well-motivated compared to other options.

- Is the inequality in Eq. 5 correct? I think one should be reversed. Is it not true that when the budget is higher the loss is lower in general?

- I do not fully understand the definition of a subnet based on its operators (in L125). I see the motivation, but the notation seems off. Why use summation for each operator within a layer but union across layers? Also, what do you mean by saying the sum of all indicator functions in a layer equals to 1? Please clarify this and better represent it in the paper.

[Post-rebuttal]
- Authors have clarified my questions and I am  satisfied with their answers.

**Limitations:**

No major negative societal impact.

**Strengths And Weaknesses:**

Strengths:

- This paper presents a well-motivated idea and sound arguments in general. The selection of spatial/temporal components makes the design simple, and the search more convenient. It makes sense to vary the number of temporal components in settings w/ different number of input frames.

- The claims of the paper are thoroughly validated in the experiments. Also, the provided ablations give insights into each component in the proposed design.


Weaknesses:

- The pretraining setup that the authors follow seems to be unfair. In fact, the authors initialize the models from scratch rather than ImageNet-pretrained weights (in contrast to previous work), citing that there is a possible information leak from the pretraining data to few-shot classes. I do agree with this comment. However, the baseline methods that the authors compare against have significantly-larger budget (params/FLOPs) and they may be at a disadvantage compared to the proposed method, if not pretrained on a large-enough dataset. Because, large models tend to overfit more on small training datasets. It would be interesting to see the performance w/o any pretraining at a similar compute budget. In Table 7, authors pretrain on UCF101 which is still a smaller dataset.

- The proposed Prototype Alignment module is well-motivated and a main contribution of the paper. However, I wonder if this is truely novel. It has similarities with the proposal in TRX which also presents a query-specific class prototype for computing distance. Also, the authors call this module 'spatio-temporal alignment' (Section 3.4), when in fact, there is no spatial information at the input to the module. This only performs temporal alignment.

- Some information missing regarding the shrinkage strategy (can be addressed with another equation). It says that an operator is discarded if its score 'deviates' from the expected score at a similar budget. The score could be significantly lower/higher in both directions beyond the threshold, and a lower score should be better as far as I can see. Is it not the case? This should be clarified.

- Writing in the paper is sometimes missing critical information and hard to follow. It can be improved. Make sure the paper is self-contained (eg: L113: 'as in [15]').

- Specific hyperparameter settings are missing, especially in the shrinkage algorithm, which is probably important to include/discuss.


[Post-rebuttal]
- Authors have responded to all my concerns and I am  satisfied with their answers.

---

> ### Author Response · Authors · 2022-08-02
> **Responses to Reviewer #1 (Part 1)**
>
> Thanks for your positive support and constructive opinion. We will enhance our writing, and the following answers will be revised accordingly in the final version.
>
> Q1: It would be interesting to see the performance w/o any pretraining at a similar compute budget.
>
> A1: Thanks for your valuable advice. We have supplemented relevant experiments in rebuttal.
> First, we compared the performance of our method and the previous methods under the condition of similar compute budget on UCF101 dataset. We increase the number of searchable layers of our model to three times of the original version stage by stage, making its Params and FLOPs similar to that of TRX. Then we evaluate the performance of extended model in 5-way 5-shot setting. All methods here use 8 frames as input. As shown in Table 13, the extended model still surpasses TRX and TimeSformer and is even slightly better than the original version. This verifies that our method is still better than the previous method at similar compute budget. Increasing the scale of the proposed model does not lead to the drop of performance, which proves that it is feasible to train such a large-scale model on this dataset.
>
> |                Method              |      Acc    |     Params    |      FLOPs    |
> |:----------------------------------:|:-----------:|:-------------:|:-------------:|
> |           TimeSformer [2]          |     63.0    |      40.7M    |     73.35G    |
> |               TRX [28]             |     67.0    |      25.6M    |     41.43G    |
> |                 Ours               |     69.7    |      8.84M    |     13.76G    |
> |     Ours (Searchable layers x3)    |     70.2    |      26.1M    |     41.19G    |
>
> In addition, we also have relevant plans for the pretraining experiments on large-scale datasets that other reviewers are concerned about. We report the performance of our model after pretraining on large-scale Kinetics-400 dataset. Similarly, all models here use 8 frames as input. Through pretraining on Kinetics dataset, all models are compared under a fair pretraining condition. After that, the few-shot dataset UCF101 is used for fine-tuning and testing, and the results are shown in the following table. As shown in Table 12, our proposed model still has the best performance after pretraining on large-scale dataset. With only 1/3 FLOPs, our method can surpass TRX by 2.5% and 10.8% on the UCF101 and HMDB51 dataset, respectively.
>
> |     Method             |     Pretraining     |             |     UCF101    |               |             |     HMDB51    |               |
> |------------------------|---------------------|-------------|---------------|---------------|-------------|---------------|---------------|
> |                        |                     |     Acc     |     Params    |     FLOPs     |     Acc     |     Params    |     FLOPs     |
> |     TimeSformer [2]    |     -               |     63.0    |     40.7M     |     73.35G    |     41.7    |     40.7M     |     73.35G    |
> |     TimeSformer [2]    |     Kinetics-400    |     80.5    |     40.7M     |     73.35G    |     54.2    |     40.7M     |     73.35G    |
> |     TRX [28]           |     -               |     67.0    |     25.6M     |     41.43G    |     46.4    |     25.6M     |     41.43G    |
> |     TRX [28]           |     Kinetics-400    |     85.1    |     25.6M     |     41.43G    |     60.7    |     25.6M     |     41.43G    |
> |     Ours               |     -               |     69.7    |     8.84M     |     13.76G    |     60.4    |     8.91M     |     13.65G    |
> |     Ours               |     Kinetics-400    |     87.6    |     8.73M     |     13.61G    |     71.5    |     8.75M     |     13.52G    |
>
> Due to page limitations of manuscript, we add the above two sets of experiments to the Supplementary Materials, from Line 624 to Line 638. We are very sorry for these two sets of late experiments.

---

> > ### Comment · Reviewer_kc2E · 2022-08-07
> > **Authors have addressed my comments on evaluation settings**
> >
> > I thank the authors for taking time to run the requested experiments. I am satisfied with these results. Regarding the final version, I suggest authors to find a way to include pretrained results in the main paper (maybe in the same table). Having the ablations under the same budget in the supplementary would be okay.

---

> > > ### Author Response · Authors · 2022-08-07
> > > **Responses to Reviewer #1 (kc2E)**
> > >
> > > Thanks for your valuable advice. According to your suggestions, we have added the pretrained results to Section 4.3.4 in the main paper, from Line 301 to Line 307.

---

> ### Author Response · Authors · 2022-08-02
> **Responses to Reviewer #1 (Part 2)**
>
> Q2: The proposed Prototype Alignment module is well-motivated and a main contribution of the paper. However, I wonder if this is truely novel. It has similarities with the proposal in TRX which also presents a query-specific class prototype for computing distance.
>
> A2: Thanks for your advice. We politely disagree that our idea is not novel compared with TRX for the following three reasons.
>
> First, our video representation method is more reasonable and efficient. TRX constructs exhaustive pairs/triplets of sampled spatial features to generate the video representations. However, our searched model naturally generates frame-level spatio-temporal representation, which already contains rich temporal information, so the laborious combination operation is omitted. The disadvantage of TRX method is obvious: the complexity will increase rapidly with the increase of the number of frames. Here we list a table to compare the differences between two generated prototypes when dealing with different input frame numbers. It can be seen from the table that the number of combination and the size of attention map will increase rapidly with the increase of the number of input frames, which is not convenient for the possible application of longer videos in the future.
>
> |     # Input Frames    |     TRX [28]    |      TRX [28]    |            Ours           |
> |:---------------------:|:---------------:|:----------------:|:-------------------------:|
> |                       |      #Pairs     |     #Triplets    |     Temporal Dimension    |
> |            4          |         6       |         4        |              4            |
> |            6          |        15       |         20       |              6            |
> |            8          |        28       |         56       |              8            |
> |           12          |        66       |        220       |             12            |
> |           16          |        120      |        560       |             16            |
>
> Second, proposed spatio-temporal prototype alignment is a nonparametric and concise solution suitable for few-shot video understanding, which can generate high-quality frame-level attention map without relying on complex subnetworks. TRX relies on a CrossTransformer subnetwork to perform alignment operations.
>
> Third, our proposed method greatly alleviates the difficulty of long video modeling, and makes it possible to align and match two complex actions. The pairs/triplets based methods have limited video understanding ability for complex actions (some complex human actions cannot be represented by only 2 or 3 sparse sampled frames). A simple example is that pairs/triplets-based approaches cannot distinguish whether a person hits the desk 3 times or 4 times. Because the maximum number of sampling frames is only 3, and it is difficult for the model to understand this repetitive action beyond triplet.
>
> Q3: Also, the authors call this module 'spatio-temporal alignment' (Section 3.4), when in fact, there is no spatial information at the input to the module. This only performs temporal alignment.
>
> A3: The reason why we named the proposed module ‘spatio-temporal prototype alignment’ is that the objective of the alignment operation is the spatio-temporal representation. Our model does generate a frame-level attention map with a dimension of TxT in the inference process, rather than spatial alignment. We have clarified relevant statements in the revised manuscript to avoid ambiguity for readers.
>
> Q4: Some information missing regarding the shrinkage strategy.
>
> A4: Thanks for your advice. In our shrinking strategy, the score of each operation is calculated according to Eq. 6. The main body of the score is based on the loss function, so the greater the loss, the worse the operation, that is, the greater the score, the worse the operation that should be discarded. We also made corresponding modifications in the manuscript to promote the understanding of relevant explanations, and updated the ‘B. Details of Transformer Space Shrinking’ subsection in Supplementary Materials. Thank you for your helpful suggestions once again.
>
> Q5: Writing in the paper is sometimes missing critical information and hard to follow. It can be improved. Make sure the paper is self-contained (eg: L113: 'as in [15]').
>
> A5: Thanks for your advice. We provide more detailed description for the ‘Spatial Downsampling Block’ in the revised manuscript. We added the content “To avoid the model being overly complex, We incorporate the dimension downsample block, named ''Spatial Downsampling Block (SDB)'' in Table 1 as in [15]. The core operation of this block is to use the down sampling in spatial self-attention, which maps an input tensor of size $(C,W,H )$ to an output tensor of size $\left(C^{\prime}, W / 2, H / 2\right)$ with $C^{\prime}>C$. Due to the change in scale, we can easily control the model complexity in different stages.” to the revised manuscript from Line 116 to Line 120.

---

> > ### Comment · Reviewer_kc2E · 2022-08-07
> > **Satisfied with the author responses, with minor comments**
> >
> > Thanks for the clarifications.
> >
> > - I think it would be meaningful to include the differences with TRX alignment (mentioned above), in the main paper. It would help a reader to choose between TRX and the proposed method.
> >
> > - I still do not agree with the term 'spatio-temporal alignment'. Although the input representation is spatio-temporal, the alignment process itself is only temporal. Renaming the term would be better than adding a clarification, but either would be okay.

---

> > > ### Author Response · Authors · 2022-08-07
> > > **Responses to Reviewer #1 (kc2E)**
> > >
> > > Q1: I think it would be meaningful to include the differences with TRX alignment (mentioned above), in the main paper. It would help a reader to choose between TRX and the proposed method.
> > >
> > > A1: Thanks for your constructive advice. We have added relevant expressions to Section 4.3.3 in the main paper, from Line 273 to Line 286.
> > >
> > > Q2: I still do not agree with the term 'spatio-temporal alignment'. Although the input representation is spatio-temporal, the alignment process itself is only temporal. Renaming the term would be better than adding a clarification, but either would be okay.
> > >
> > > A2: Thanks for your valuable advice. According to your suggestion, we have renamed this term “Temporal Alignment (TA)” to avoid confusion. And we have updated all relevant descriptions in the main paper and supplementary materials. We hope that our improved manuscript can address your concerns.

---

> ### Author Response · Authors · 2022-08-02
> **Responses to Reviewer #1 (Part 3)**
>
> Q6: Specific hyperparameter settings are missing, especially in the shrinkage algorithm, which is probably important to include/discuss.
>
> A6: Thanks for your advice. To implement the shrinking strategy, we conduct an operation mask for all operations to control the selectivity of supernet for operations. The initial value of the mask is set to 1, indicating that the operation can be selected. During the training, the loss value and structure of each subnet will be recorded. We calculate the score of all candidate operations for every two epochs according to Eq. 6. Then, all operations are ranked according to their scores, and the top 10% of operations are discarded. The mask values corresponding to the discarded operations will be set to 0, indicating that these operations are in an unselectable state in the subsequent search. After that, the following search process can be carried out more efficiently on this shrunk space.
> We have made corresponding modifications in the manuscript to promote the understanding of relevant explanations, and updated the ‘B. Details of Transformer Space Shrinking’ subsection in Supplementary Materials
>
> Q7: Why is this the best search space? This is not well-motivated. I believe the authors should discuss why these dimensions (spatial/temporal/heads) are the ones to search in and whether it is sufficient. There are many other aspects such as spatio-temporal resolution, channel-expansion, parallel/multi-path connections, components beyond self-attention. I agree that including all these axes will be practically impossible, but the choices of authors should be well-motivated compared to other options.
>
> A7: Thanks for your advice. As you state, there are massive operations and hyper-parameters for the model to choose from. In the process of designing search space, we drew from many prior excellent works. First, in terms of the overall structure of the model, [13] proposes that the video understanding model has different emphasis on the resolution of time and feature maps in different stages. And the manually designed X3D model  has achieved great success in video understanding task. Few-shot action recognition places high demands on the ability of the video representation, which motivated us to utilize NAS to explore the model structure. We hope our model can spontaneously choose and focus on different types of information at different stages to obtain better representations. Second, in terms of video understanding through NAS, [21] and [41] explored the method of searching 3DCNN, and both achieved good performance. Considering the natural advantages of Transformer in sequence analysis, we plan to design search space based on Transformer. Third, through the comparison of various space-time modules, the manually designed TimeSformer confirms the effectiveness of the Divided Space-Time Attention module. Finally, we extract independent Space ''SAB'' and Time Attention Blocks ''TAB'' to build the final search space. It is true that there are still some axes that have not been explored in this work. How to find the optimal network architecture in a super large search space is also a challenging task. We are also willing to make further experiments and attempts in this larger field in the future.
>
> According to your comments, we have added a new subsection named ‘K. Motivations for Our Search Space’ to Supplementary Materials. Thank you for your constructive suggestions once again.
>
> Q8: Is the inequality in Eq. 5 correct? I think one should be reversed. Is it not true that when the budget is higher the loss is lower in general?
>
> A8: Thanks for your advice. This inequality is indeed reversed. As we replied in A6, the score of each operation is calculated according to Eq. 6. The main body of the score is based on the loss function, so the greater the score, the worse the operation. This is a mistake in our description. Our overall experiment still follows the correct understanding. We have modified Eq. 5 in revised manuscript. Thanks again for your patience here.
>
> Q9: I do not fully understand the definition of a subnet based on its operators (in L125). Please clarify this and better represent it in the paper.
>
> A9: Thanks for your advice. We use $\bigcup_{i} \sum_{j} 1_{j}^{i} O_{i, j}$  to represent the subnet $a$ because our method selects operators layer by layer. In each layer of supernet, subnet selects only one operation. Thus, for the indicator function $1_{j}^{i} \in\{0,1\}$ in $i$-th layer, the sum of the indicator functions is 1: $\sum_{j} 1_{j}^{i}=1$. Finally, the symbol $∪$ indicates that the selected operations of each layer are combined to form a complete network. We have modified related expressions in revised manuscript, from Line 133 to Line 135.

---

> > ### Comment · Reviewer_kc2E · 2022-08-07
> > **Not satisfied with response A7, others are okay**
> >
> > The clarification provided by authors in A7 does not answer my question Q7. I agree that in the mentioned prior work it has shown the benefits of considering different spatio-temporal resolutions, divided space-time attention and so on. In your search space, selection of spatial and temporal components is reasonable. Among other options for search (eg: spatio-temporal resolution, channel-expansion, parallel/multi-path connections, components beyond self-attention), why do you select number of heads as the only other dimension of search? Why this is the optimal (or reasonable-with-budget) search space is still not well-motivated.

---

> > > ### Author Response · Authors · 2022-08-07
> > > **Responses to Reviewer #1 (kc2E)**
> > >
> > >  Thanks for your valuable advice. Our initial design goal is that the search space can accommodate both video understanding and feature extraction ability. And selecting the # head is mainly based on the consideration of search complexity. We hope to answer the reasons from the following folds:
> > >
> > >
> > > (1) The actual performance of NAS is determined by the space complexity and search efficiency. Considering too many dimensions may affect the performance of the searched model. In some previous Transformer NAS methods (e.g. AutoFormer [6], ViTAS [33]), it is mentioned that # head and channel dimension are indeed important for model design. But comparatively speaking, the search for # head is not easy to lead to huge search space. This is one of its advantages. In this work, the size of our search space is $4.74 \times {10} ^{18}$. If we follow the setting of ViTAS [33] to search the token embedding dimension, the search space may grows to $4.74 \times {10} ^{42}$. If we follow the setting of AutoFormer [6] to search the embedding dimension and MLP ratio (the ratio of hidden dimension to the embedding dimension in the multi-layer perceptron), the search space will become larger (even $6.32 \times {10} ^{51}$). Therefore, we finally referred to the settings of TimeSformer [2] in the channel dimension without conducting channel search.
> > >
> > >
> > > (2) For spatio-temporal resolution, we divide the model into three stages with different resolutions, giving the model some freedom to choose spatio-temporal resolution. We also found that, when fewer input frames are fed in, more temporal attention blocks will be placed in the third stage to improve the action modeling capability.
> > >
> > >
> > > (3) For parallel/multi-path connections and components beyond self-attention, these are indeed not considered in this work. Because the search space in this paper is mainly inspired by Transformer-based video understanding and image classification models. Our original intention is to design a relatively simple and smart search space.
> > >
> > > Thanks again for your patience here. Although some of the axes you mentioned are not directly studied in this paper, it is an exciting direction for us. Searching for channel search, parallel/multi-path connections and more other operations are expected to build a more unified and diverse search space and further excavate the potential of NAS in neural network design. We really hope to do this in our future work.

---

> > > > ### Comment · Reviewer_kc2E · 2022-08-07
> > > > **Satisfied with the answer, please include a discussion in supplementary**
> > > >
> > > > Thanks for stating the details on how search space will grow. I agree that there needs to be a compromise of selected axes. Please add this discussion to supplementary and put things in perspective.

---

> > > > > ### Author Response · Authors · 2022-08-07
> > > > > **Responses to Reviewer #1 (kc2E)**
> > > > >
> > > > > Thank you for your support. I will add the relevant contents to the supplementary later. Thanks again for your patience.

---

> ### Author Response · Authors · 2022-08-06
> **Further discussion with reviewer kc2E**
>
> Dear reviewer kc2E:
>
> We thank you for the precious review time and valuable comments. We have provided corresponding responses and results, which we believe have covered your concerns. We hope to further discuss with you whether or not your concerns have been addressed. Please let us know if you still have any unclear parts of our work.
>
> Best,

---

### Meta-Review · Area_Chair_TKdS · 2022-08-30

**Recommendation:** Accept
**Confidence:** Certain

**Metareview:**

All three reviewers lean towards the acceptance of the paper. The reviewers believe the rebuttal has addressed their concerns. The AC recommends acceptance of the paper, and suggest the authors to include the materials and the discussion they promised in the rebuttal in the final version of the paper.

**Award:**

No

---

### Decision · Program_Chairs · 2022-09-14

Accept